# Sustaining Our Relationship: Dyadic Interactions Supported by Technology for People with Dementia and Their Informal Caregivers

**DOI:** 10.3390/ijerph191710956

**Published:** 2022-09-02

**Authors:** Viktoria Hoel, Eliva Atieno Ambugo, Karin Wolf-Ostermann

**Affiliations:** 1Institute for Public Health and Nursing Research, University of Bremen, 28359 Bremen, Germany; 2Leibniz Science Campus Digital Public Health, 28359 Bremen, Germany; 3Department of Health, Social and Welfare Studies, Faculty of Health and Social Sciences, University of South-Eastern Norway (USN), 3616 Kongsberg, Norway

**Keywords:** dementia, caregiving, psychosocial interventions, dyadic relationships, social technology, mobile health apps, user needs, usability

## Abstract

Impaired memory function and challenges in communication affect the ability of people living with dementia to interact with family caregivers socially. The onset of dementia in a family member and the communication challenges that follow can lead to conflict, isolation and loss of closeness in the relationship. I-CARE is a tablet-based technology providing leisure activities specifically designed for people living with dementia to do in tandem with caregivers. The intention is that caregiving dyads engage with I-CARE together, using the activities contained in the system as the basis for positive social interactions. This paper reports on a mixed-methods feasibility study of I-CARE, evaluating the system’s usability and assessing the impact on caregiving dyads. We also explored barriers and facilitators to independent use of the technology among community-dwelling people living with dementia and their family caregiver. Results suggest that I-CARE is a feasible tool to facilitate positive experiences in dementia caregiving dyads. Important relationship outcomes for the participating dyads were enrichment in social interactions, facilitated communication, having a shared activity and relationship sustenance. Successful uptake requires continuous proactive support tailored to the needs and preconditions of users over an extended time until they feel confident using the system independently.

## 1. Introduction

Dementia is a term describing a collection of symptoms that affect cognitive functions, such as impaired memory and orientation, and challenges in communication capabilities [1]. The onset of dementia in a family member and the communication challenges that follow can lead to conflict, loss of closeness and isolation [2,3]. Family caregivers are not experts in communicating with people living with dementia (PLWD) [4] and might struggle to find effective ways to engage their loved ones with dementia in social interactions and conversations [5].

Families, particularly couples affected by dementia, have to adjust to the transition from an interdependent relationship to a dyadic caregiving relationship of caregiver-care recipient [1,6,7]. This transition can negatively impact the relationship [8], influencing the well-being of both dyad members [9,10,11,12] and increasing the risk of unwanted institutionalisation [13,14]. Conversely, research indicates that supportive relationships that validate positive attributes of the individual can enable PLWD to live meaningful lives while preserving their personhood [11,15,16,17]. There is growing evidence to suggest that psychosocial interventions promoting social relationships can enhance the well-being of PLWD and their caregivers alike [11,18,19]. Nonetheless, there are few psychosocial interventions aiming specifically to sustain dyadic relationships in the context of dementia [20,21].

Psychosocial interventions are relational in nature [17], with advocates of person-centred care articulating the benefits for dementia caregiving dyads focusing on ways to create mutually satisfying relationships that support well-being and quality of life (QoL) [17,22]. Moreover, interventions focusing on improving or maintaining social relationships in dementia can have beneficial effects on clinical and social outcomes for PLWD (such as improved mood and cognitive function, and reduced use of antipsychotics) [11,23], as well as helping caregivers in their role (such as stress management and learning how to support PLWD) [24,25]. Greater attention should therefore be directed towards research and development of psychosocial interventions to sustain dyadic relationships in dementia.

An increasing area of interest in psychosocial interventions in dementia is digital technologies [26,27]. However, progress in technological interventions supporting relationships through interaction and communication for this population is still in its infancy [9,28,29]. For this study, we utilise the term ’social technology’, defined as “any technology that facilitates social interactions and influences social processes between people” [30] (p. 3). Despite their novelty, psychosocial interventions using social technology have shown positive effects on both dyad members and positive relationship gains, including reduced caregiver burden and increased relationship quality (as rated by the dyad members) [9,31]. This calls for further exploration of the feasibility and potential benefits of utilising social technology to facilitate positive interactions between PLWD and their caregivers and ultimately support the dyadic relationship. In this context, the current study evaluated I-CARE, a tablet-based activation system explicitly designed for PLWD to do in tandem with a caregiver.

This study aimed to evaluate the usability of the I-CARE system in terms of usefulness and user-friendliness to inform a future, large-scale trial. Furthermore, the impact of the technology-supported dyadic social sessions on the participating caregiving dyads and their caregiving relationship was assessed. Finally, the present study explores barriers and facilitators to implementing I-CARE, including as pertains to the ability of community-dwelling dementia caregiving dyads to use the system independently.

## 2. Methods

### 2.1. Study Design

A mixed-methods approach [32] was conducted with a single group pre/post-test with follow-up design to examine the impact of a tablet-based intervention conducted in the homes of PLWD and family caregivers. The present study sought to assess a dyadic psychosocial intervention in a home-based environment and how tablet-facilitated social activities might provide active engagement, thus triggering positive experiences in caregiving, which could be explored in future studies.

In developing this dyadic intervention, we drew from and built upon the theoretical perspective of Cartwright and colleagues’ model of enrichment processes in caregiving (EPC) [33]. This theoretical framework conceptualises enriching events in family caregiving of frail older adults [33]. Cartwright et al. explicate how some families use pleasurable or meaningful experiences in their caregiving roles and define enrichment as “the process of endowing caregiving with meaning or pleasure for both caregiver and care recipient” [33] (p. 32). By facilitating engaging dyadic activities, the present study aimed to support enriching experiences in caregiving through shared moments that are meaningful to both caregiver and care recipient.

### 2.2. The I-CARE System

I-CARE is a tablet-based, user-specific activation system specifically designed to actively engage PLWD in social interactions in tandem with a caregiver [3,34]. The content encompassed in I-CARE comprises a wide array of activities, including image galleries, classical music, videos, short stories, proverbs, quizzes and games. The activities have different complexities and difficulty levels, making I-CARE suitable for most ages, types and stages of dementia. What makes I-CARE unique is that the system tracks feedback to estimate which content is most successful in activating the user. This information feeds into a dynamic recommender system that tailors the content to the participant as it learns about the user.

In the current study, the system was installed on Huawei Media Pad M5 (10.8-inch display) tablets and placed on a table in front of the dyads. At the beginning of any I-CARE session, the system asks about the daily well-being (“How are you today?”) of the PLWD, using a smiley rating scale (positive, neutral, negative). After this assessment, the recommender system suggests four activities based on user information (e.g., age, (previous) occupation, interests) and previous ratings by the user if available. The dyad can also choose content using the system’s search function or activity history. After each activity, the system asks the PLWD to rate how they liked the activity, again using a smiley rating scale. Thereafter, the system returns to the content overview with new recommendations. Dyads can decide at any point whether to continue with another activity or cancel an ongoing one.

Seven project partners from science, industry and the social service sectors collaborated on developing I-CARE as a tablet-based system that supports the individual needs and potential of PLWD [35,36]. In the systems development phase, the active involvement of participants in the design and selection of activation content was a central guiding principle. Therefore, technically operationalisable guidelines were defined in cooperation with gerontology scientists, field experts, and industrial partners. A complete account of the content and development of I-CARE can be found elsewhere [35].

### 2.3. Participants and Procedure

From December 2020 to December 2021, caregiving dyads were recruited through local health and care organisations, support groups and information centres in and around Bremen, Germany. The study included nine community-dwelling PLWD and their family caregiver. People with a formally confirmed diagnosis of dementia were included, regardless of the type or stage, while the caregiver (age ≥ 18) had to either cohabitate with the PLWD or visit at least twice per week on average. Recruitment and inclusion criteria are further elaborated elsewhere [3].

This study received ethical approval from The Deutsche Gesellschaft für Pflegewissenschaft e.V (DGP), and informed consent was obtained from all participants. Before enrolling, each dyad received a tablet with the I-CARE system installed. They also had training and time to familiarise themselves with the user platform together with a research assistant. Moreover, each tablet was equipped with short tutorials explaining every aspect of the system (content, searching) and the tablet (turning on/off, charging). Finally, the participants were instructed to use I-CARE together at least three times per week for at least 15 min (if possible). Each dyad participated in the I-CARE intervention for four weeks.

### 2.4. Data Collection

Quantitative: In this feasibility study, the objective was partly to evaluate the impacts of the dyadic activities encompassed in the I-CARE system. The primary outcome measures included caregiver burden for the family caregiver, quality of life for both dyad members, and their relationship quality. Standardised, internationally validated instruments were utilised and administered at three points in time for each dyad recruited; at baseline, post-intervention and four-week follow-up, and included measurement instruments outlined in Table 1.

Participant characteristics were also collected, including age, gender, education, number of children, and number of caregiving hours the family caregiver provided. Although a formal diagnosis of dementia was an inclusion criterion, the severity of functional disability associated with cognitive impairment among the participants with dementia might influence the usability of I-CARE. The Functional Assessment Staging questionnaire (FAST; [37]) was therefore also employed at baseline to get a sense of the stage of dementia in the participating dyads. FAST is a functional scale designed to allow caregivers to chart the decline of people diagnosed with dementia in seven functional disability stages, each assessing clinical descriptions of common abilities, such as memory capabilities, personal hygiene, and taking care of oneself (e.g., dressing or eating). This hierarchy of functions has been found to be related to dementia-related cognitive decline [38]. The assessment was done by the family caregiver utilising FAST as a proxy rating of the care recipient’s functioning and might therefore differ from ratings by caregivers with a medical background (e.g., nurse or general practitioner).

Qualitative: Finally, semi-structured interviews were conducted post-intervention to explore further the impact of I-CARE sessions and the usability and possible barriers and facilitators of using I-CARE independently at home. Baseline semi-structured interviews were also conducted, focusing on the impact of the COVID-19 pandemic on participants’ use of social technology in general and their adjustments to the caregiving role. The findings of the baseline interviews are reported elsewhere [3]. The interviews—lasting between 45 min to three hours—were conducted in the participants’ homes by a trained research assistant, audio-recorded and transcribed verbatim. An important aspect of the post-intervention interviews was to inquire whether both dyad members found I-CARE useful or experienced usability issues. Usability includes determining the extent to which specified users can use a product to achieve specified goals with effectiveness, efficiency, and satisfaction in a specified context [48]. Guided by the work of Lund (2001), usability is assessed in terms of usefulness and user-friendliness. Usefulness concerns whether users believe an application fulfils specific needs, and user-friendliness refers to whether users believe an application will be easy and simple to use [49]. A translated excerpt of the interview guide containing questions related to usability issues is available in the Appendix A.

### 2.5. Data Analysis

Descriptive statistics were used to describe the characteristics of the participants and average differences in the reported outcomes between baseline (t0) and post-intervention intervention (t1); and between post-intervention (t1) and at four-week follow-up (t2) were calculated. A paired t-test was used to assess the differences in scores between baseline and post-intervention and between post-intervention and follow-up. Due to the small sample size, *p*-values were approached in an exploratory manner to identify which outcomes may be worth investigating in a future, large-scale study; *p*-values < 0.05, therefore, do not infer statistical significance. Statistical analysis was performed using Stata software (Stata©. Stata Corp., College Station, TX, USA version: 15).

The dyadic interviews were transcribed verbatim and translated from German to English by a native German speaker. The transcripts were subjected to inductive, data-driven thematic analysis following the steps outlined by Braun and Clarke [50]. The translated transcripts were read multiple times to get familiar with the data. Thereafter, the first and second authors (VH and EAA) independently coded features of the data before collating the codes and initial themes. Themes were then discussed, reviewed, and revised at the level of the individually coded extracts and the complete data set by sorting codes, transferring codes under similar sub-themes, and collapsing or creating new themes. The main themes and subthemes were refined and named collaboratively. Finally, themes were discussed between all authors, achieving a consensus. Questions relating to the usability of I-CARE were collaboratively analysed and organised in terms of participants’ reflections on usefulness and user-friendliness. The software NVivo version 12 (QSR International Pty Ltd., Melbourne, Australia, 2020) was used to facilitate the data’s systematic organisation and analysis.

## 3. Results

### 3.1. Quantitative: Descriptive Characteristics of the Participants

Eighteen participants (nine PLWD and nine family caregivers) enrolled in the I-CARE intervention, all but one dyad (two sisters) had a spousal relationship. Table 2 provides an overview of the sample’s demographic information. All care recipients had a formal diagnosis of dementia, with a FAST score ranging from mild to severe. The mean age was 77 for PLWD (range 58–89; SD 9.47) and 72 for caregivers (range 57–87; SD 12.04). Seventy-eight per cent of the care recipients were male, while 89% of the caregivers were female. There was only one male caregiver in the sample. The average amount of caregiving activities provided by the caregiver lay at nine hours per day (range 2–16; SD 6.07). One dyad withdrew halfway into the intervention due to the lack of motivation in the husband with dementia. Another dyad could not participate in the follow-up due to the rapidly deteriorating health of the husband with dementia, who moved into a nursing home shortly thereafter.

### 3.2. Qualitative: I-CARE Usability

The I-CARE interface was developed in collaboration with PLWD and caregivers during the systems development phase and is meant for use in the community and an institutional setting without special training or expertise [35,36]. Nevertheless, the usability of I-CARE for community-dwelling PLWD and their family caregiver has not been explicitly assessed, let alone in the context of a global pandemic where in-person technical support is limited. Hence, the usability of I-CARE was explored through post-intervention semi-structured interviews in terms of usefulness and user-friendliness.

#### 3.2.1. Usefulness

Of the nine dyads who completed the intervention, 78% (7/9) of the participants found I-CARE useful, while the husbands with dementia in two dyads (Dyad 7 and Dyad 9) did not find I-CARE useful and were uninterested when engaging with I-CARE. The husband in Dyad 7 did not enjoy playing games and did not see the purpose of the other activities (such as the image gallery and music activities), while the husband in Dyad 9 generally did not enjoy using technology and instead preferred more haptic activities such as working in the garden. Touchscreen-based activities were therefore not suitable for his needs or interests. The issue in these two cases seemed to be that the activities were contained in digital devices. Although I-CARE is specifically designed for PLWD, the importance of the human element was a recurring topic. Unprompted, several participants mentioned that it was not necessarily the specific activities but rather the conversations and shared moments that arose that were enriching. The specific ways in which I-CARE generated these enriching experiences are elaborated further below under ‘Identified themes’.

#### 3.2.2. User-Friendliness

On average, the dyads engaged with I-CARE 3.4 times per week, while the average duration of each session was around 22 min. I-CARE was found to be accommodating and flexible to the dyads: every I-CARE session could be easily initiated, and any activity could be concluded according to the preferences of the dyads without having to click through lengthy procedures. An important feature of the I-CARE system is the varying difficulty levels in several activities that make the system flexible to fit users’ different cognitive levels. Nevertheless, the findings indicated that some participants found the activities too complex for their partner with severe dementia, while caregivers of partners with mild dementia found the activities too simple. Examining the interface’s user-friendliness showed that although the dyads mostly managed to use the system independently, not all participants could do so effortlessly. The main challenge in these cases was technical with regard to navigating to find content outside the system’s recommendation page, such as specific activities they had done in previous sessions. Moreover, some dyads struggled with technical difficulties. Participants occasionally experienced activities freezing or being inaccessible. Although these errors were temporary, they required working with the back-ended system, making it impossible for the participants to solve technical issues themselves. This gave rise to another serious problem, as technical issues were not always reported immediately.

### 3.3. Quantitative: Primary Outcome Measures

Most of the participants could answer the questionnaires without difficulties and had no reservations in answering questions they considered personal. Several participants commented that the questions made them think about aspects of their caregiving relationship upon which they had not previously reflected. Although all *p*-values in the present study are exploratory, a ‘Last Observation Carried Forward’ (LOCF) analysis was used to account for the two dyads who did not complete the follow-up assessments (Dyad 1 and Dyad 9). As shown in Table 3, QoL in the PLWD (DEMQOL/DEMQOL-Proxy) decreased by −0.67 points throughout the study, but the decrease seemed larger in the follow-up period without the I-CARE sessions (observed cases (OC) and LOCF decreasing by 9.57 and 7.4 points, respectively). This tendency can also be seen in the quality of the caregiving relationship (QCPR), as rated by the PLWD. There was a non-significant 4.75-point increase in overall QCPR during the intervention, with a significant decrease in the follow-up period (OC: −3.5; LOCF: −2.63). When looking at the warmth/affection subscale, a significant 3.25-point increase was found post-intervention for PLWD, followed by a non-significant decrease in the follow-up period (OC: −1.33; LOCF: −1.0).

PLWD consistently reported higher values for relationship quality than the caregivers (see also Figure 1). Both dyad members seemed to have experienced relationship gains during the intervention, although this increase was non-significant. No differences in outcomes were found for the family caregivers post-intervention. During the four-week follow-up (t2), a significant decrease in the rating in the total quality of the dyadic relationship could be observed (OC: −3.0; LOCF: −2.33). This was also the case for the warmth/affection subscale (OC: −2.0; LOCF: −1.56), but this finding was borderline insignificant when missing values were imputed using the LOCF. There seemed to have been a larger decrease in caregivers’ QoL (CarerQol-7D) during the I-CARE intervention compared to the follow-up period. This trend was, however, non-significant. Finally, the perceived caregiver burden (BSFC) seemed to have a non-significant 3.67-point decrease during the I-CARE intervention before increasing in the follow-up period (OC: 1.29; LOCF: 1.0).

### 3.4. Qualitative: Identified Themes

Two main themes were identified when analysing the transcripts from the semi-structured interviews reporting the experiences and reflections of the I-CARE facilitated social sessions among the dyads: “The beneficial effects on the dyadic relationship” and “Technology requirements”. Although the subthemes within these two were sufficiently distinguishable to remain as individual themes, they were nevertheless closely connected in that they mutually influenced each other. Table 4 provides an overview of the themes, subthemes and example quotes.

#### 3.4.1. The Beneficial Effects on the Dyadic Relationship

The participating dyads reported several positive experiences in their caregiving relationship while using the I-CARE system to facilitate social sessions. The emerging patterns in the dyads’ reported experiences led to four distinct subthemes: (i) enrichment in social interactions; (ii) facilitating communication; (iii) providing a shared activity; and (iv) togetherness in the relationship.

Enrichment in social interactions: First and foremost, I-CARE was perceived as a meaningful activity that actively stimulated the partner with dementia to participate in the social sessions. For some caregivers, this stimulation was a welcomed change in the dyadic interactions, as some struggled to find appropriate activities that engaged their loved ones since the onset of dementia. I-CARE provided rewarding sessions for the caregiver, who found that active engagement of their partner with dementia facilitated positive experiences in their caregiving relationship. Several caregivers also reported being positively surprised by their partner’s capabilities in engaging with the system, such as answering quizzes, solving puzzles or navigating through the activities. Learning trivia or history from their partner with dementia was also a positive experience arising from using I-CARE.

Facilitating communication: I-CARE was perceived as a helpful conversation aid, as the sessions stimulated exchanges within the dyad. As they explored I-CARE, figuring out how to operate the system and navigating through the activities sparked discussions between the care recipient and caregiver. The activities also introduced new topics in the dyadic conversations, revolving around historical events or old memories. Thus, the communication between dyad members was also closely connected to reminiscing and discussing distinct or shared memories from the past.

Although I-CARE supported topic introduction and sparked discussions, caregivers also reported non-verbal communication. Despite their impaired communication abilities, participants with more severe dementia responded non-verbally to the caregiver’s comments on the I-CARE activities and content. I-CARE thus supports the dyadic relationship by helping introduce topics, facilitating non-verbal social interaction.

Providing a shared activity: By serving as a point of joint attention, I-CARE provided dyads with something they could do together. Some caregivers described how it was becoming increasingly difficult to find engaging activities accommodating a dementia diagnosis, and I-CARE was a welcomed new activity to explore together. The system was designed to be easy to use by a tandem partner without special training or expertise. This coincided with the general impression of the dyads, who found I-CARE accommodating, with sessions being simple to initiate when they felt motivated. Nevertheless, operating the I-CARE system was at times challenging for some caregivers who had never owned a tablet before or were unfamiliar with touchscreen technology. However, this unfamiliarity provided a sense of mastery for some, both care recipients and caregivers. Several dyads described the joy they experienced when they operated the tablet or navigated the novel device.

Sustaining togetherness: ‘Together’ was a key term permeating the discussions around the benefits of using I-CARE, leading to the identification of the fourth theme. The conversations that arose while using I-CARE, and being jointly engaged and having fun together contributed to a sense of sustained togetherness in the dyadic relationship. Several caregivers reported having struggled to sustain this feeling in the course of their loved one’s illness. For participants without experience with similar tablet-based activities, I-CARE was also an opportunity to explore something new together. Doing something entirely different from their everyday routine was perceived as valuable. The interactive nature of the activities was perceived as meaningful to the dyad members, offering more opportunities to be actively engaged as a pair compared to, e.g., watching television in silence. The participants’ experiences were in harmony with the vision of I-CARE, a system that was developed not to offer cognitive training or respite for caregivers but rather to offer activities that are meaningful in and of themselves [34,36].

#### 3.4.2. Technology Requirements

The participants’ perceived needs to use I-CARE independently at home were central to the post-intervention interviews. It quickly became apparent that multiple prerequisites and user needs must be met to experience the beneficial effects of social technology in a dementia caregiving context. The second overarching theme identified was, therefore, “Technology requirements”, with two subthemes described below: (i) Barriers to overcome; and (ii) Facilitators to promote.

Barriers to overcome: When looking into the barriers that were perceived or experienced by the dyads, the patterns revealed a clear distinction between user-related barriers and barriers related to I-CARE in general. As the barriers related to I-CARE were identified when exploring its usability, the emphasis in this section lies on the user-related barriers to using social technology for community-dwelling PLWD and their family caregivers. These barriers involved different considerations depending on whether the participant’s role in the dyad was that of a caregiver or a care recipient. For the PLWD, the barriers were mainly related to their cognitive capabilities to engage with I-CARE. Although I-CARE is specifically designed to cognitively and socially activate PLWD [35], some caregivers felt their partner’s dementia was too advanced to fully engage in the I-CARE sessions. This capability was nevertheless partly dependent on the PLWD’s capacity on the given day, which dictated his/her level of interest and motivation, consequently influencing the level of encouragement needed from the caregiver.

In most dyads living with an advanced stage of dementia, caregivers more frequently reported the intervention as tiresome, requiring greater energy to take on a supporting role. The social sessions with I-CARE were perceived as an additional task for caregivers already pressed on time and energy caring for their family member with dementia. The caregivers’ capacity to engage with I-CARE was also heavily influenced by their tech literacy. Some participants had never owned a tablet before, let alone led tablet-based social activities such as those available in I-CARE. Consequently, several participants emphasised the need for knowledge and experience with social technology in general to ease the use of novel technology like I-CARE.

Facilitators to promote: Nevertheless, a major contributor to the successful use of I-CARE among participants was the tech interest, which seemed to mitigate the limited tech literacy. Interest in exploring new activities was a clear facilitator among both dyads with prior experience with social technology and those with barely or none. Their curiosity and willingness to put themselves into unfamiliar territory were essential to experiencing the positive discoveries the dyads made together. One remarkable example was Dyad 4. Even though they had no prior experience in social technology, to the research team’s surprise, they figured out how to make a profile picture for their I-CARE profile, yet doing so had not been demonstrated to them when they received their tablet. As mentioned, this discovery gave both dyad members a sense of mastery. Despite their lack of experience with similar technology, the dyad’s enthusiasm stood in contrast to other dyads with extensive technical skills but who were not equally interested in the I-CARE system.

The user-related barriers in the use of technology were further accentuated by the identified facilitator encompassing the need for close and continuous support. The research team had emphasised that they would be available to support the dyads whenever needed, regardless of the dyads’ questions, issues or timing. Still, none of the dyads reached out for additional support unless it was related to a technical issue. Additionally, the tablet’s home page was equipped with several tutorials because navigating the I-CARE system could be difficult for some dyads despite the training sessions. These comprised two-minute videos with step-by-step guides to navigate the system and operate the tablet. However, when interviewing the dyads, it quickly became apparent that none of them had used the videos for additional support.

## 4. Discussion

This mixed-methods feasibility study aimed to assess the usability of I-CARE and evaluate the system’s impact on dementia caregiving dyads and their relationship. With the small number of existing publications reporting on psychosocial interventions targeting dyadic relationships in dementia caregiving [20]—fewer still supported by technology —our findings contribute to filling the existing knowledge gap on how social technology might support relationship sustenance between PLWD and their caregivers, ultimately improving clinical and social outcomes for both dyad members.

We also explored the participating dyads’ perceptions of barriers and facilitators to using I-CARE independently together at home. By identifying contextual factors influencing the implementation of novel technology in the homes of PLWD and their caregivers, future research can build on our findings to overcome barriers and promote facilitators. Overall, the results suggest that the I-CARE intervention is a feasible intervention for community-dwelling dementia caregiving dyads that has the potential to positively influence the caregiving relationship by engaging PLWD and their family caregivers in meaningful activities together. We wish to emphasise that the pandemic context must be considered when interpreting all aspects of our findings, including the impacts of using I-CARE when other social leisure activities were heavily restricted, as well as usability issues and barriers to independent use of I-CARE with limited in-person tech support available.

### 4.1. Usability Issues

The usability of I-CARE was explored by investigating the dyads’ opinions of the usefulness and user-friendliness of the activation system. The findings show that I-CARE was perceived as useful to all participants except for the PLWD in two dyads. Here, the main issue seemed to be the technical nature of the activities. In both these cases, the wives displayed disappointment in their husbands’ lack of motivation to do the activities—as they had enjoyed using I-CARE and generally struggled with finding other activities they could do together since the onset of dementia. Social technology aiming to facilitate social interactions needs to meet the dyad as a unit, with content adjusted for both dyad members.

Nevertheless, the individual personalities and the compatibility of the dyad members’ interests will influence their engagement in a specific activity. I-CARE was perceived as useful to the remaining dyads by bringing them together around a meaningful activity. We argue that any activity or device (technological or otherwise) that can join the caregiver and care recipient together in positive social interactions could benefit the caregiving relationship. Although not useful to all, I-CARE seems to be a feasible tool for generating enriching experiences in caregiving dyads by serving as a point of joint attention. This is concurrent with existing research on technology-supported dyadic activities, which shows how technology can be a crutch or a tool to facilitate positive social interactions in dementia caregiving dyads [9,51,52,53].

Most participants found the system user-friendly in that it was easy to initiate and end the sessions without lengthy procedures. The feedback on the activities’ difficulty levels was conflicting; some dyads living with a more severe dementia stage found some I-CARE activities too difficult, while those with a mild case found the activities too simple. There seems to be room to expand I-CARE’s content and difficulty levels to suit PLWD with varying levels of cognition. When examining the ease with which users could navigate the system, participants reported that they could mostly do so independently, with technical errors being the reason behind any confusion and frustration. Such errors were a major concern to the study team, considering that the participants were operating I-CARE in the privacy of their own homes. As such, technical issues could be solved only when the participants reported them. However, participants rarely reported issues when they occurred and instead waited until the next scheduled meeting or check-up session with a research team member. When asked why they had not immediately called, they responded that they did not want to “bother” the researchers and would rather wait for the next scheduled meeting to bring up issues. This coincides with research pointing to generational differences in seeking support [54], with older adults being more reserved in asking for technical assistance than the younger generations—reasons including being embarrassed by their lack of technical knowledge, not understanding the terminologies used, or not wishing to inconvenience others [54,55,56,57]. To mitigate this, the research team scheduled two extra visits with each dyad during the intervention phase to ensure that the technology functioned and answered any questions the participants had. However, a high frequency of home visits in a large-scale trial might be challenging, and any technical errors in the back-end system must be solved before future trials.

### 4.2. Quantitative Outcomes

Reported quality of life (QoL) among the PLWD showed a negative trend throughout the study, though this decline was less steep during the intervention period. Although the sample size of nine caregiving dyads is too small to infer statistical significance, QoL and rated quality of the caregiving relationship nevertheless seemed to be positively influenced by the intervention. The findings thus suggest that I-CARE is in harmony with similar research, including tablet-based interventions for dementia caregiving dyads, that report positive impacts on both dyad members and their caregiving relationship [31,58,59,60,61].

Similar to the PLWD, a steady decrease in the QoL of caregivers was observed during the study. This is consistent with research showing that informal caregivers’ quality of life steadily decreases as the severity of the care recipient’s dementia progresses [62,63,64]. However, a slightly steeper decrease in the QoL of caregivers was observed during the intervention period compared to the follow-up without the I-CARE sessions. Although this trend was non-significant, this might be explained by the added burden of being a tandem partner; using novel technology or being confronted by the consequences of dementia on their family member’s cognitive abilities or social functioning. These factors might have been augmented by the ongoing pandemic and the related social restrictions. More research is needed to investigate whether the decline is caused by the fact that the I-CARE activities were contained in novel technology and whether there is an interaction effect therein with the severity of the PLWD’s dementia. There would then be a need to intervene accordingly if, for example, the demands of using novel technology are shown to influence caregivers’ QoL negatively. The decline in QoL was, however, not reflected in the reported relationship quality nor the caregiver burden outcome. The latter showed a larger though non-significant reduction during the intervention than during the follow-up period. The validity of CarerQol-7D in evaluating QoL of caregivers of PLWD has been demonstrated [44], and the instrument is broadly used in dementia research [43,65,66,67]. Even so, the fact that the instrument contains seven questions, with only one encompassing the caregiving relationship, might limit the instrument’s sensitivity to small changes in interpersonal aspects influential to caregivers’ well-being. Further assessment of the CarerQol-7D instrument is needed in more extensive trials focusing on caregiving relationships. The non-significant statistical tests of the quantitative measures should not be interpreted as indicative of poor feasibility of future planned research. It is argued that the outcomes of feasibility studies should be measured with descriptive statistics and qualitative analysis rather than null hypothesis testing [68].

### 4.3. Qualitatively Exploring the Impacts of I-CARE 

Evaluating psychosocial interventions targeting caregiving dyads as a unit constitutes a complex research problem, calling “for answers beyond simple numbers in a quantitative sense, or words in a qualitative sense” [69] (p. 21). Using a mixed-methods approach comprising qualitative and quantitative measures, we gained in-depth information about participants’ experiences and opinions. The qualitative component expatiates the positive trends observed in the quantitative outcomes measuring the dyads’ relationship quality. Two overarching themes were established, related to the primary outcomes of I-CARE and prerequisites for generating positive experiences using the I-CARE system. Within the first overarching theme, ‘the beneficial effects on the dyadic relationship’, four subthemes were identified: (i) enrichment in social interactions; (ii) facilitating communication; (iii) providing a shared activity; and (iv) togetherness in the relationship. The dyads’ positive experiences arising from engaging with I-CARE together are promising indications of the potential of technology to support relationship sustenance in a dementia caregiving context. Similar themes have been observed in the body of literature exploring how technology might support dyadic interactions in a dementia caregiving setting; in our preceding systematic literature review, we found that technologies could support dyadic social interactions through four main mechanisms: (i) breaking the ice by initiating dialogue and serving as a conversational platform; (ii) increasing interaction frequency and duration by encouraging more involvement between the dyad members; (iii) understanding the PLWD better through reminiscence activities; and (iv) reducing pressure on the caregiver by making the communication more reciprocal [9].

Although the usability assessments of I-CARE were primarily positive, the second overarching theme from the qualitative component shows that certain prerequisites must be met to generate positive outcomes from technology-driven dyadic interventions. One significant barrier to successfully engaging with tablet-based activities seemed to be the stage of dementia and capability on a given day, directly influencing how much support and energy the caregivers had to invest in fulfilling their role as tandem partners. Caregivers who had difficulty motivating their partner with dementia were more inclined to view I-CARE as an additional task than an opportunity to connect. This perception was also influenced by caregivers’ familiarity with technology, although the importance of tech literacy was inconsistent among participants. Some caregivers considered it vital to have previous experience before exploring novel technology, while others emphasised that being interested in trying novel technology was more decisive. Participants who expressed joy from exploring something entirely new and unfamiliar seemed more likely to view I-CARE as an enriching activity. The contending emphasis on tech literacy versus tech interest is congruent to participants’ reflections before enrolling in our study, where we found that challenges related to tech literacy, to a certain extent, were mitigated by user willingness [3]. This contributes to the growing evidence contradicting the existing misconception of older adults being unable or unwilling to follow technological development [70,71,72].

Regardless of whether one ranks having experience with social technology or a general interest in it as most vital, the need for proactive and continuous support for end-users in learning novel technology seems indisputable. Although the dyads mostly managed to operate I-CARE independently, some still reported struggling with navigating through the system and finding specific content. Despite a comprehensive training session, a recommender system adapting content based on the user’s interests, and video tutorials showing how to navigate the system, the dyads reported occasionally getting lost or confused. Nevertheless, none of the participants reached out for assistance unless a technical error occurred, and even then, they did not always call for support. The aversion to seeking help, either for guidance or technical assistance, clearly shows that support for this population cannot be passive, with users being expected to reach out themselves. The non-use of tutorial videos and few requests for help indicate that support for using novel technology should be provided to this population proactively and preferably in person. This is especially important in the familiarisation phase until end users—especially caregivers—gain confidence in using the novel technology independently.

### 4.4. Limitations

Although there were positive trends in the primary outcome measures, our results can be ambiguous or biased due to the small sample size such that statistical significance cannot be inferred. Thus, caution must be exercised in drawing conclusions about the potential impact of I-CARE. The COVID-19 pandemic posed a major barrier to recruitment and data collection, severely limiting the generalizability of results. Furthermore, the results can be influenced by selection bias. The majority of the dyad members with dementia were male, and all but one caregiver were female. In addition, the relationship type was spousal in all dyads but one. There is a lack of knowledge about the heterogeneity of dyadic relationship constellations in home-based dementia care settings [21], making it difficult to determine whether our participants differ from a nationally representative sample. Nevertheless, our sample share many of the same characteristics as that of the German DemNet-D study [21,73,74], in terms of age and gender distribution [74], dementia severity [73], living situation [74] and relationship typology (couple relationship, the wife being the caregiver) [21].

Some of the participants with more severe stages of dementia were limited in the extent to which they could provide comprehensive descriptions of their own experiences and reflections on I-CARE. The caregivers, therefore, provided most of the information during the in-depth interviews. Despite the interviewer and caregiver consistently asking whether the dyad member with dementia agreed with the caregiver’s statement or wanted to make further comments to the discussion, the PLWD participated less such that the caregivers’ views heavily influenced the qualitative component. Supplementing audio recordings and field notes with video observations in the moment of storytelling might be a necessary strategy to accommodate the person with dementia to express their thoughts and experiences in cases where the dementia stage complicates active participation in interviews. Finally, to adhere to COVID-19 restrictions, interviews were conducted using protective masks and adhering to social distancing guidelines. A desire to reduce dyads’ and interviewers’ exposure to one another might have limited the length of the interviews, thereby reducing the richness of the data.

## 5. Conclusions

The results from our study suggest that I-CARE is a feasible tool to facilitate enriching experiences in dementia caregiving dyads. Important relationship outcomes for the participating dyads were enrichment in social interactions, facilitated communication, having a shared activity and relationship sustenance. Our qualitative research methods enabled us to explore and elaborate on outcomes such as those we have identified here, which standardised instruments can be limited in assessing. For the quantitative component, first positive results have been observed, but further research with larger sample size and a control group is needed to assess the effectiveness of I-CARE and provide statistically robust results. This study should be replicated with a control arm comprised of similar activities as I-CARE but not contained in a digital device (such as board games, printed photos, or storybooks) to support the establishment of the value (or the lack thereof) of utilising social technology to facilitate dyadic interactions in a dementia caregiving context. More research is also needed to establish whether the decline in caregivers’ QoL stems from taking on the role of a tandem partner using novel technology and whether this is correlated to the severity of dementia in their family member. Regardless, I-CARE’s design and back-end system need to be revised before further implementing and assessing the activation system. To ensure the successful uptake of I-CARE and similar systems in a home-based environment, it is crucial to have continuous proactive technical support tailored to the needs and preconditions of users (including the severity of dementia of the care recipient) over an extended period until they feel confident in using the system independently. Including a facilitator might help lighten any extra burden caregivers face in their tandem partner role, allowing them to take on an equal role in the I-CARE activities. This could foster the incorporation of I-CARE for daily use among community-dwelling caregiving dyads. Moreover, with sufficient development and technical support, social technologies such as I-CARE could be implemented on a large scale as an interaction aid for PLWD and their caregiver in institutional settings, regardless of whether the caregiver is a visiting family member or a member of staff.

## Figures and Tables

**Figure 1 ijerph-19-10956-f001:**
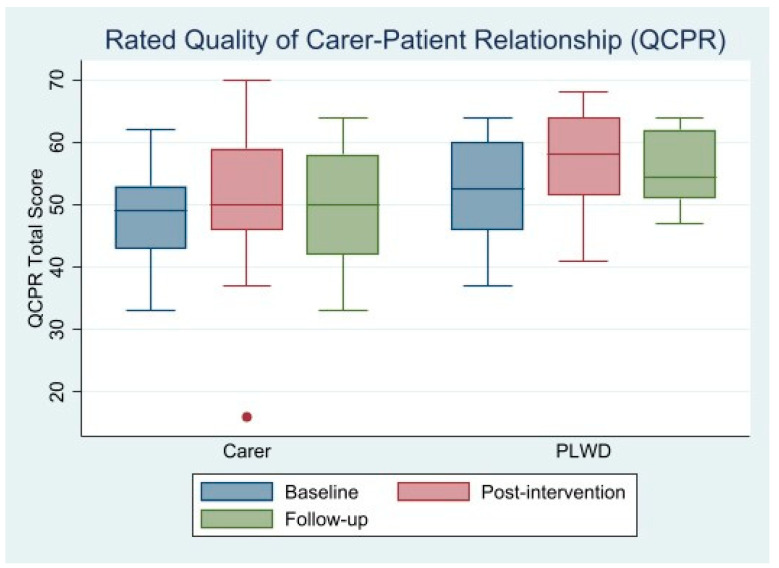
Boxplot summarising the rated quality of the caregiving relationship, as measured by the QCPR instrument.

**Table 1 ijerph-19-10956-t001:** Outcome/screening measures.

Outcome	Tool	Description
**Person living with dementia**		
**Dementia severity**	Functional Assessment Staging Tool (FAST) [37]	18-item functional scale designed to allow caregivers to chart the decline of people diagnosed with dementia on seven functional disability stages. Range: 1–7f. A higher score indicates more severe impairment (1 = no impairment up to 7 = very severe impairment. Has been found to be related to cognitive decline in dementia [38].
**Quality of Life**	Dementia Quality of Life (DEMQOL/DEMQOL-Proxy) [39]	28/31(Proxy) item tool designed to measure Health-Related Quality of Life (HRQoL) in PLWD. Addresses five domains: cognition, negative emotion, positive emotion, social relationships, and loneliness. 4-point Likert scale, ranging from “a lot” (1p), “quite a bit”, “some”, “not at all” (4p). The scale includes an additional global QoL item (item 29 resp. 32). Range: DEMQOL: 28–112; DEMQOL-Proxy: 31–124. A higher score indicates better HRQoL. The original and German versions have shown good construct validity and internal consistency [39,40,41].
**Informal caregiver**		
**Quality of Life**	Carer Quality of Life (Carer-Qol-7D) [42]	7-item tool combined with Visual Analogue Scale (VAS) designed to measure the impacts of providing informal care. Addresses two positive and five negative dimensions of informal caregiving. 3-point Likert scale, ranging from “no” (0p), “a little” to “a lot” (2p) for positive dimensions, inverted for the negative dimensions. Range: A weighted sum score is calculated from these items, ranging from 0 to 100 [43]. A higher score indicates a higher quality of life. The convergent and clinical validity to evaluate the impact of providing informal care for people with dementia has been demonstrated [44].
**Caregiver Burden**	Burden Scale for Family Caregiving (BSFC) [45]	28-item tool designed to measure subjective burden of family caregivers. 4-point Likert scale, ranging from “strongly agree” (0p) to “strongly disagree” (3p), with 12 items inversely presented. Range: 0–84. A higher score indicates a higher caregiver burden. BSFC sum score assigned to three Subjective Burden Categories: mild (0–35p), moderate (36–45p) and severe (46–84p). Represents a comprehensive concept of caregiver burden, and has been found to allow for valid assessments in both research and practice [46].
**Dyad**		
**Relationship Quality**	Quality of Caregiver-Patient Relationship (QCPR) [47]	14-item tool designed to assess the quality of the caregiving relationship. Contains two subscales assessing warmth/affection and absence of conflict/criticism in the caregiving relationship. 5-point Likert scale, ranging from “totally disagree” (1p) to “totally agree” (5 p) with a neutral option (“undecided” (3p)). The conflict/criticism subscale is scored inversely. Range: 14–70. A higher score indicates a higher quality of the relationship.

**Table 2 ijerph-19-10956-t002:** Demographic characteristics of participants (N = 9 caregiving dyads).

PERSON LIVING WITH DEMENTIA				CAREGIVER				DYADIC RELATIONSHIP			
ID	Gender: Age	FAST ^a^	Education	ID	Gender: Age	Caregiving Hours ^b^	Education	Type	Cohabiting	Children	Services ^d^
P1	M: 58	Severe	Secondary	C1	F: 57	11–13 h	Secondary	Spousal	**✓**	**✓**	None
P2	M: 80	Mild	Tertiary	C2	F: 70	2–3 h	Secondary	Spousal	**✓**	**✓**	Day Centre
P3	M: 85	Moderate–severe	Secondary	C3	F: 83	>14 h	Secondary	Spousal	**✓**		Home assistant
P4	F: 82	Severe	Secondary	C4	M: 86	9–10 h	Secondary	Spousal	**✓**	**✓**	Day Centre
P5	M: 69	Mild–moderate	Tertiary	C5	F: 58	2–3 h	Tertiary	Spousal	**✓**	**✓✓ ^c^**	None
P6	M: 89	Severe	Secondary	C6	F: 87	>14 h	Secondary	Spousal	**✓**	**✓**	Assisted living
P7	M: 83	Severe	Tertiary	C7	F: 76	4–6 h	Primary	Spousal		**✓**	Assisted living
P8	F: 77	Severe	Secondary	C8	F: 75	>14 h	Secondary	Siblings	**✓**		None
P9	M: 73	Mild	Secondary	C9	F: 59	2–3 h	Tertiary	Spousal	**✓**	**✓✓**	Home Assistant

^a^ Functional Assessment Staging Tool; ^b^ Average number of hours per day spent on caregiving activities by caregiver; ^c^ Children that are minors, still living at home; ^d^ Respite or home services.

**Table 3 ijerph-19-10956-t003:** Average differences in outcomes from baseline (t0) to post-intervention (t1), and from post-intervention to follow-up (t2).

	POST-INTERVENTION			FOLLOW-UP					
	Observed Cases (OC)			Observed Cases (OC)			Last Observation Carried Forward (LOCF)		
	AD	Paired *t*-Test (*p*-Value)	95 % CI	AD	Paired *t*-Test (*p*-Value)	95 % CI	AD	Paired *t*-Test (*p*-Value)	95 % CI
**Person living with dementia (PLWD)**									
*DEMQOL/* *DEMQOL-Proxy*	–0.67	–0.206 (0.842)	[–8.11, 6.78]	–9.57	–3.126 **(0.020) ***	[–17.06, –2.08]	–7.40	–2.728 **(0.026) ***	[–13.73, –1.15]
*QCPR Total*	4.75	1.129 (0.296)	[–5.20, 14.70]	–3.5	–5.218 **(0.003) ***	[–5.22, –1.78]	–2.63	–3.479 **(0.010) ***	[–4.41, –0.84]
*QCPR Warmth*	3.25	2.542 **(0.038) ***	[0.23, 6.27]	–1.33	–1.754 (0.140)	[–3.29, 0.62]	–1.00	–1.673 (0.138)	[–2.41, 0.41]
*QCPR Criticism*	–2.13	–0.653 (0.534)	[–9.82, 5.57]	2.17	2.291 (0.071)	[–0.26, 4.60]	1.63	2.089 (0.075)	[–0.21, 3.46]
**Caregiver**									
*CarerQol-7D*	–3.83	–0.744 (0.478)	[–15.72, 8.05]	–2.60	–0.973 (0.368)	[–9.14, 3.94]	–2.02	–0.974 (0.359)	[–6.81, 2.77]
*QCPR Total*	1.11	0.254 (0.806)	[–8.97, 11.19]	–3.00	–3.240 **(0.018) ***	[–5.27, –0.73]	–2.33	–2.800 **(0.023) ***	[–4.25, –0.41]
*QCPR Warmth*	0.22	0.102 (0.921)	[–4.79, 5.23]	–2.00	–2.449 **(0.049) ***	[–3.99, –0.00]	–1.56	–2.256 (0.054)	[–3.15, 0.03]
*QCPR Criticism*	–0.89	–0.392 (0.706)	[–6.12, 4.34]	1.00	1.620 (0.156)	[–0.51, 2.51]	0.78	1.575 (0.154)	[–0.36, 1.91]
*BSFC*	–3.67	–1.103 (0.302)	[–11.33, 4.00]	1.29	0.881 (0.412)	[–2.28, 4.86]	1.00	0.885 (0.402)	[–1.61, 3.61]

AD = Average difference; BSFC = Burden Scale for Family Caregiving; CarerQol = Care-Related Quality of Life; DEMQOL = Dementia Quality of Life; OC = Observed Cases; LOFC = Last Observation Carried Forward; QCPR = Quality of Carer-Patient Relationship. * Significant at 5% level. Note: N = 18 (nine PLWD and nine caregivers) at t0 and t1, while N = 14 (seven PLWD and seven caregivers) at t2.

**Table 4 ijerph-19-10956-t004:** Themes, subthemes, collated codes and example quotes from the post-intervention dyadic interviews.

Themes	Subthemes	Example Quotes
**The beneficial effects on the dyadic relationship**	Enrichment in social interactions	“Yes, it was rewarding. Worth it or not. In what way should it be worth? It was interesting to see, what my husband is interested in and what can we accomplish together. We enjoyed it, that also is a goal.” (C2)
	Facilitating communication	“When we were doing this, we eventually sat there for more than half an hour and watched, and [my husband] started to talk about different topics, that were connected to the pictures in some way. These topics were old memories of friends and so on. I thought that was good, very, very good. That was worth it. Just for that, it was worth it.” (C5) “No, he simply watched that and he thought it was very nice […] for example the cat, what they are up to. I could see he was interested. But talking about it…he had forgotten it a moment later. That’s how it is. What happened five minutes ago, is instantly gone again. So, we can’t talk together. We just [hesitates] *are* together, when I visit.” (C6)
	Providing a shared activity	“It was fun, and gave us something new to talk about and explore together.” (C3) “Like we used it now, yes absolutely. It was a nice new routine to have together, instead of just watching TV silently. We could just whip up the tablet instead when we wanted.” (C8)
	Togetherness in the relationship	“There are hours where she sits in her chair and simply looks out of the window, without saying anything or even standing up, for hours! So, when we are using this device and try to figure it out it’s much more interesting to be together. This half an hour or 15 min or longer, where we are fiddling around with the device, is much more useful, for me as well because it is this “togetherness” that we create.” (C4)
**Technology requirements**	Barriers to overcome	“Both need to be a bit interested in technology, or at least one of them have to know how to fix it if something is wrong, or to know where to get help.” (C1) “[…] how we were together, that was fun! I was surprised that it worked so well to get her interested. Every time we managed to use the thing, it was an adventure [laughs]. That was also a positive surprise when we managed to take the [I-CARE profile] picture. We are delighted every time when we see the picture of ourselves [on the tablet].” (C4)
	Facilitators to promote	“Well, that there’s good instructions and continuous support. If you have any questions concerning the technology, like you said that some people struggle to use it, it’s important to have someone in the background who can help out. No matter when or what you wonder about.” (C9) “Joy doing it. That you have fun doing it. I think that’s the most important thing and if it’s not enjoyable, it’s not possible.” (C7)

## Data Availability

The data presented in this study are not publicly available due to privacy regulations. A translated excerpt of the interview guide containing the question items related to I-CARE Usability is available as Appendix A.

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
