# Peer review of "Sustaining Our Relationship: Dyadic Interactions Supported by Technology for People with Dementia and Their Informal Caregivers"

_ijerph, 2022, doi:10.3390/ijerph191710956_

Round 1
Reviewer 1 Report
Thank you for the opportunity to review this novel pilot study that investigated the feasibility of the I-CARE app to support people living with dementia and their caregivers.
You have provided an incredibly detailed, well-structured report that clearly outlines the potential benefits of this technology whilst also continuing to consider factors that can be complicating when working with this population.
Two minor changes to be noted:
- Table 2 – is it possible to include specific type of dementia diagnosis ?
- Page 7 line 236, you mentioned musical activities however earlier in the description of the content included in the app this was not mentioned – can you please include earlier on.
Finally, you have raised the issue of dementia severity creating increased burden on caregivers engaging with this app, which is an important point to consider, particularly moving forward. As such, I think it would be worthwhile to bring this point back into the conclusion.
Congratulations on an important contribution to dementia care.
Author Response
- English language and style: English language and style are fine/minor spell check required
- Response: A rigorous spellcheck has been performed, which should raise the language throughout the manuscript.
- Checklist: No improvement points
- Response: All referrals to ‘Pages’ and ‘Lines’ in our comments are directed at the version with changes tracked. Please note: When reading the manuscript in MS Word, lines/pages may differ from what has been referred to in the responses to the reviewer, depending on which version of Word is being used. Using the PDF version to review the revision is advised.
- Thank you for the opportunity to review this novel pilot study that investigated the feasibility of the I-CARE app to support people living with dementia and their caregivers. You have provided an incredibly detailed, well-structured report that clearly outlines the potential benefits of this technology whilst also continuing to consider factors that can be complicating when working with this population. Two minor changes to be noted:
- Response: Thank you for engaging with our manuscript and for your kind words coupled with important suggestions to raise the quality of our paper. Our answers and reactions to your comments are as follows:
- Table 2 – is it possible to include specific type of dementia diagnosis?
- Response: As the sample is from a small community, we did not include the specific types of dementia diagnosis in this article, in order to preserve the anonymity of the participants. We hope that this is acceptable, as we did not exclude any type or stage of dementia in our eligibility criteria.
- Page 7 line 236, you mentioned musical activities however earlier in the description of the content included in the app this was not mentioned – can you please include earlier on.
- Response: Thank you for pointing out this oversight. The musical content has now been included (Page 3, line 107)
- Finally, you have raised the issue of dementia severity creating increased burden on caregivers engaging with this app, which is an important point to consider, particularly moving forward. As such, I think it would be worthwhile to bring this point back into the conclusion.
- Response: Thank you for pointing out this important consideration. We have included your suggestion in the discussion and conclusion (Page 16, line 537-542, page 18, line 658-661 and line 664-668).
- Congratulations on an important contribution to dementia care.
- Response: Thank you for your kind words.
Reviewer 2 Report
Dear authors,
Thank you for this comprehensive and well presented article. Here are my comments:
- Introduction, page 2, lines 51-54 : explain why it would be important to study and develop psychosocial interventions to sustaing dyadic relationships in the context of dementia.
- You use the acronym PLWD for people living with dementia on the first page, but for example, on page 2, line 76, you spell it out again. It needs to be harmonised.
- Discussion, page 14, lines 499-512 : it would be relevant to add the information that technological tools should not add burden to the carer in dyadic use or in supervision of the use by the person with dementia.
- Discussion, page 15, lines 523-535 : This paragraph would benefit from parallels with elements of the literature to see if similar themes are observed.
- Conclusion : thought should be given to asking the question of the same activity but without the technological support to see if the interest is the same or not as well as the adherence to the programme.
In addition, the question of whether or not a person was facilitating the programme would reduce the burden on the carer who would no longer be in the position of carer but a partner in the activity.
Elements of openness and questions of daily or even large-scale application would be welcome in conclusion.
Best wishes
Author Response
- Dear authors, Thank you for this comprehensive and well-presented article. Here are my comments:
- Response: Thank you for engaging with our manuscript and for your critical appraisal, lifting the overall quality of this paper. All referrals to ‘Pages’ and ‘Lines’ in our comments are directed at the version with changes tracked. Please note: When reading the manuscript in MS Word, lines/pages may differ from what has been referred to in the responses to the reviewer, depending on which version of Word is being used. Using the PDF version to review the revision is advised. Our answers and reactions to your comments are as follows:
- Introduction, page 2, lines 51-54: explain why it would be important to study and develop psychosocial interventions to sustaining dyadic relationships in the context of dementia.
- Response: Thank you for pointing out this indistinctness. We have elaborated our arguments and changed the order of some of them to make the importance of psychosocial interventions to sustain dyadic relationships clearer (Page 2, line 49-63).
- You use the acronym PLWD for people living with dementia on the first page, but for example, on page 2, line 76, you spell it out again. It needs to be harmonised.
- Response: Thank you for making us aware of this inconsistency. We have now gone through the manuscript to ensure that PLWD is used in harmony (Page 2, line 89 and page 4, Table 1).
- Discussion, page 14, lines 499-512: it would be relevant to add the information that technological tools should not add burden to the carer in dyadic use or in supervision of the use by the person with dementia.
- Response: Thank you for bringing our attention to this important consideration. We have included your suggestion in the discussion section (Page 16, line 537-542).
- Discussion, page 15, lines 523-535: This paragraph would benefit from parallels with elements of the literature to see if similar themes are observed.
- Response: Thank you for pointing out this weakness in our discussion. We have now drawn parallels to the body of literature in this paragraph (Page 16, line 570-577).
- Conclusion: thought should be given to asking the question of the same activity but without the technological support to see if the interest is the same or not as well as the adherence to the programme
- Response: Thank you for raising this issue. We have included this reflection in our conclusion (Page 18, line 654-658).
- In addition, the question of whether or not a person was facilitating the programme would reduce the burden on the carer who would no longer be in the position of carer but a partner in the activity.
- Response: Thank you for pointing out this important consideration. We have included your suggestion in the conclusion (Page 18, line 658-661 and line 666-668).
- Elements of openness and questions of daily or even large-scale application would be welcome in the conclusion.
- Response: Thank you for the recommendation. We have now broadened the conclusion to include some reflections on this (Page 18, line 668-673).
Reviewer 3 Report
1. Generally, this is an interesting paper. The authors discussed important implications for using "social technology" to sustain the relationship between persons living with dementia and their caregiving dyads and improve their quality of life.
However, I am mostly concerned with the limited sample size in this study. As Rausch et al. (2017) mentioned, limited sample size and lack of statistical power in quantitative outcomes may lead to unclear results or risk of bias. Acknowledging this in the limitation section and reporting 95% confidence intervals may strengthen this paper.
There are some other suggestions. The authors could consider moving some details to the Appendix (e.g., Table 2) so that the main text is more focused.
At the meantime, the authors might need to provide more details about the semi-structured interviews. For example, how the questions related to "usefulness" were asked, and the average length of each semi-structured interview.
The authors also need to explicitly explain their originality. It is not very clear how they filled current literature gaps. If they made any methodological innovations (e.g., innovations in measuring “usefulness”), they should point it out.
2. Line 16. Is I-CARE a full-name or an abbreviation?
3. Line 42. Institutionalisation may not necessarily lead to worse outcomes. The authors might actually refer to "unwanted institutionalisation" or "low-value institutionlisation".
4. Line 52: Cite “Rausch, A., Caljouw, M. A., & van der Ploeg, E. S. (2017). Keeping the person with dementia and the informal caregiver together: a systematic review of psychosocial interventions. International Psychogeriatrics, 29(4), 583-593”.
5. Line 141: What are the lengths of the baseline and post-intervention periods?
6.Table 2. Since the sample size is small, it is important to understand more demographic characteristics in the sample and learn about how different/similar this sample is compared to a nationally representative sample. For example, the authors could report socioeconomic status such as education (or income) in the sample. If there is no such information, the authors should acknowledge this as a limitation.
7. Line 275: report specific magnitudes of decline in QoL with and without I-CARE sessions.
8. Line 279-280: Report specific numbers.
9. Table 3. Report 95% confidence intervals in the main table or in the Appendix.
Reference: Rausch, A., Caljouw, M. A., & van der Ploeg, E. S. (2017). Keeping the person with dementia and the informal caregiver together: a systematic review of psychosocial interventions. International Psychogeriatrics, 29(4), 583-593
Author Response
- English language and style: English language and style are fine/minor spell check required
- Response: A rigorous spellcheck has been performed, which should raise the language throughout the manuscript.
Checklist:
- Response: The authors hope that the adjustments, elaborations and justifications are satisfactory in order to meet the criteria of the checklist.
1a. Generally, this is an interesting paper. The authors discussed important implications for using "social technology" to sustain the relationship between persons living with dementia and their caregiving dyads and improve their quality of life.
- Response: Thank you for taking the time to engage with our manuscript, for which your critical review contributed to improving the reporting quality. All referrals to ‘Pages’ and ‘Lines’ in our comments are directed at the version with changes tracked. Please note: When reading the manuscript in MS Word, lines/pages may differ from what has been referred to in the responses to the reviewer, depending on which version of Word is being used. Using the PDF version to review the revision is advised.
1b. However, I am mostly concerned with the limited sample size in this study. As Rausch et al. (2017) mentioned, limited sample size and lack of statistical power in quantitative outcomes may lead to unclear results or risk of bias. Acknowledging this in the limitation section and reporting 95% confidence intervals may strengthen this paper.
- Response: We thank you for pointing out the risk of unclear results and/or bias. We have adjusted accordingly to make this limitation clearer (Page 17, line 614-617. To further mitigate this limitation, we have followed your suggestion and included a 95 % confidence interval in Table 3 (Page 8, line 304).
1c. There are some other suggestions. The authors could consider moving some details to the Appendix (e.g., Table 2) so that the main text is more focused.
- Response: We are grateful for the suggestion of moving Table 2 to the appendix. We hope the reviewer agrees with keeping Table 2 in the main text: as the qualitative component comprises a substantial part of our manuscript, we view the inclusion of participant characteristics in the main text as important to provide background to participants’ quotes in Table 4.
1d. At the meantime, the authors might need to provide more details about the semi-structured interviews. For example, how the questions related to "usefulness" were asked, and the average length of each semi-structured interview
- Response: Thank you for raising the issue of the insufficient description of questions related to the semi-structured interview. To address this issue, we have included information on the length of the interviews (Page 5, line 182-183). In addition, we have added a translated excerpt of the interview guide (Referred to as ‘Usability questions, interview guide excerpt’ section (Page 5, line 191-193, page 18, line 675-676 and page 19 line 695-697).
1e. The authors also need to explicitly explain their originality. It is not very clear how they filled current literature gaps. If they made any methodological innovations (e.g., innovations in measuring “usefulness”), they should point it out.
- Response: Thank you for making us aware of the lack of an explicit description of originality. We have now explicitly stated how our research might contribute to filling the existing knowledge gap on psychosocial interventions supporting dyadic relationships in dementia caregiving (Page 14, line 458-467).
- Line 16. Is I-CARE a full-name or an abbreviation?
- Response: I-CARE is the full name. The second title of the initial project of I-CARE was “Individuelle Aktivierung von Menschen mit Demenz”, which was later changed and formally named „Individual Care for People with Dementia. We have kept the name as is, to harmonise with preceding publications reporting on I-CAREs system development phase:
i) Schultz, T., et al., I-CARE-An Interaction System for the Individual Activation of People with Dementia. Geriatrics, 2021. 6(2): p. 51.
ii) Steinert, L., et al., Towards Engagement Recognition of People with Dementia in Care Settings, in Proceedings of the 2020 International Conference on Multimodal Interaction (ICMI ’20). 2020: Virtual event, Netherlands. p. 558-565.
iii) Schultz, T., et al., I-CARE - Ein Mensch-Technik Interaktionssystem zur Individuellen Aktivierung von Menschen mit Demenz, in Zukunft der Pflege: Tagungsband der Clusterkonferenz. 2018: Oldenburg
- Response: I-CARE is the full name. The second title of the initial project of I-CARE was “Individuelle Aktivierung von Menschen mit Demenz”, which was later changed and formally named „Individual Care for People with Dementia. We have kept the name as is, to harmonise with preceding publications reporting on I-CAREs system development phase:
- Line 42. Institutionalisation may not necessarily lead to worse outcomes. The authors might actually refer to "unwanted institutionalisation" or "low-value institutionlisation".
- Response: Thank you for raising this important distinction. We have incorporated your suggestion into our manuscript (Page 2, line 44).
- Line 52: Cite “Rausch, A., Caljouw, M. A., & van der Ploeg, E. S. (2017). Keeping the person with dementia and the informal caregiver together: a systematic review of psychosocial interventions. International Psychogeriatrics, 29(4), 583-593”.
- Response: Thank you for making us aware of this relevant and important article. We are happy to include this citation to strengthen our arguments (Page 2, line 51). In addressing the other reviewers’ comments, we have adjusted and restructured this paragraph (Page 2, line 49-63).
- Line 141: What are the lengths of the baseline and post-intervention periods?
- Response: Thank you for pointing out this possibly confusing description of the periods. We have now clarified that these data collections were at specific points in time points (Page 4, line 155. See also line page 3, 147-148, where the intervention period is specified).
- Table 2. Since the sample size is small, it is important to understand more demographic characteristics in the sample and learn about how different/similar this sample is compared to a nationally representative sample. For example, the authors could report socioeconomic status such as education (or income) in the sample. If there is no such information, the authors should acknowledge this as a limitation.
- Response: The authors are grateful for raising this issue. Table 2 has been updated to include the level of education among participants (Page 6, line 236). In addition, we have addressed the question of representativeness in the ‘Limitations’ section (page 17, line 621-628).
- Line 275: report specific magnitudes of decline in QoL with and without I-CARE sessions.
- Response: Specific numbers have now been included in the text (Page 8, line 293-295).
- Line 279-280: Report specific numbers
- Response: Specific numbers have now been included in the text (Page 8, lines 296-300 and Page 10, lines 319-320 and 324-326).
- Table 3. Report 95% confidence intervals in the main table or in the Appendix.
- Response: Table 3 has been revised to include 95 % confidence interval (Page 8, line 304)
- Reference: Rausch, A., Caljouw, M. A., & van der Ploeg, E. S. (2017). Keeping the person with dementia and the informal caregiver together: a systematic review of psychosocial interventions. International Psychogeriatrics, 29(4), 583-593
- Response: Citation has been included in the manuscript (Page 2, line 51). In addressing the other reviewers’ comments, we have adjusted and restructured this paragraph (Page 2, line 49-63).